# Proposal for an objective binary benchmarking framework that validates each other for comparing MCDM methods through data analytics

Mahmut Baydaş[1], Tevfik Eren[1], Željko Stević[2], Vitomir Starčević[3] and Raif Parlakkaya[4]

[1] Faculty of Applied Sciences, Necmettin Erbakan University, Konya, Turkey
[2] Faculty of Transport and Traffic Engineering, University of East Sarajevo, Doboj, Bosnia and Herzegovina
[3] Faculty of Business Economics, University of East Sarajevo, Bijeljina, Bosnia and Herzegovina
[4] Faculty of Political Sciences, Necmettin Erbakan University, Konya, Turkey



## ABSTRACT

When it comes to choosing the best option among multiple alternatives with criteria of different importance, it makes sense to use multi criteria decision making (MCDM) methods with more than 200 variations. However, because the algorithms of MCDM methods are different, they do not always produce the same best option or the same hierarchical ranking. At this point, it is important how and according to which MCDM methods will be compared, and the lack of an objective evaluation framework still continues. The mathematical robustness of the computational procedures, which are the inputs of MCDM methods, is of course important. But their output dimensions, such as their capacity to generate well-established real-life relationships and rank reversal (RR) performance, must also be taken into account. In this study, we propose for the first time two criteria that confirm each other. For this purpose, the financial performance (FP) of 140 listed manufacturing companies was calculated using nine different MCDM methods integrated with step-wise weight assessment ratio analysis (SWARA). İn the next stage, the statistical relationship between the MCDM-based FP final results and the simultaneous stock returns of the same companies in the stock market was compared. Finally, for the first time, the RR performance of MCDM methods was revealed with a statistical procedure proposed in this study. According to the findings obtained entirely through data analytics, Faire Un Choix Adéquat (FUCA) and (which is a fairly new method) the compromise ranking of alternatives from distance to ideal solution (CRADIS) were determined as the most appropriate methods by the joint agreement of both criteria.

## INTRODUCTION

The use of an accurate procedure to measure financial performance (FP) is important to improve a user's decisions about a company (*e.g.*, investor, manager, partner, supplier and creditor). Financial performance is an important indicator that reveals whether a company

Corresponding author
Željko Stević,
zeljkostevic88@yahoo.com

successfully fulfills its mission or not (*Ayhan & Önder, 2021*). It is therefore reasonable to use MCDM methods because of their success in considering the multidimensional nature of FP with precision. As a matter of fact, MCDM methods have been preferred as a selection and ranking method in many previous studies and in many different scientific fields (*Kabir & Hasin, 2013*; *Stević et al., 2022a*; *Pala, 2022*; *Gvozdović et al., 2022*; *Oral, 2021*; *Zhang et al., 2022*). Of course, a single criterion or ratio is insufficient for an overall assessment of the multidimensional performance of companies. In this sense, MCDM methods take into account the dimensions that express the different purposes of the enterprises and transform them into a single score (*Diakoulaki, Mavrotas & Papayannakis, 1995*; *Zopounidis et al., 2015*; *Rasoulzadeh & Fallah, 2020*; *Turhan & Aydemir, 2021*; *Khodadadi-Karimvand & Shirouyehzad, 2021*; *Edu, Agoyi & Agozie, 2021*; *Baydaş, 2022a*). Thus, a fair comparison becomes possible for companies.

On the other hand, due to the lack of clear consensus, making a valid decision on which MCDM method to choose when measuring companies' FP for a given problem is a challenging issue that is both intriguing and alarming. Although this may seem like an intricate and insoluble problem for methodologists, sensitivity, validation, or robustness analyzes are being developed to better understand the nature of the methods (*Triantaphyllou, 2000*). Moreover, the literature currently reports that the number of MCDM methods is more than 200 (*Danesh, Ryan & Abbasi, 2017*; *Cinelli et al., 2022*). On the other hand, a more robust, anxiety-reducing, and comforting framework can be suggested compared to a random MCDM selection. Simply put, MCDM methods can be compared based on their ability to relate to real life. This brings to mind the naturally occurring sequences in real life. The share price is an example, and similar rankings can also be used in other scientific fields. It is a known fact that many MCDM types (provided that the same decision matrix and weighting coefficient are used) produce similar rankings at certain scales (*Karaoğlan & Şahin, 2018*). However, MCDM approaches have unique algorithms and therefore MCDM methods may differ in suggesting the 'best alternative'. This situation directly affects the decisions of the decision makers. The factors that differentiate the ranking results of MCDM methods are the normalization type, assumptions, limitations and threshold value, along with different calculation procedures. In addition, this situation affects the general ranking to a certain extent. These features, which make MCDM methods unique, actually facilitate a good comparison. In the past, the most important problem for MCDM selection was the absence of an objective benchmark. However, the authors were mostly comparing calculation procedures with MCDM inputs; a character analysis based on MCDM outputs can show us a clearer path. In this sense, "price", which establishes a relationship with the FP of the companies simultaneously, can be an anchor criterion. As it is known, the share price of companies is a solid example from real life. The price is formed as a result of the consensus of thousands of stock market investors in the market. In fact, the price phenomenon is a dynamic system that is affected by many factors, similar to financial performance. In other words, although it is represented by a single score (just like MCDM scores), it consists of multiple criteria. While FP is simply a phenomenon that can be controlled by the company, the share price is a dynamic that the company cannot directly control. There is a simultaneous

relationship between these two factors in certain constraints. Moreover, the level of this existing relationship varies according to the ability of MCDM methods (*Baydaş, 2022b*). At this point, the fact that some MCDM methods produce a regular, consistent and strong relationship with the share price shows the special capacity of MCDM methods on a subject. In this study, an "output-based" solution obtained with "data analytics" is proposed as an alternative to a classical "input-based" methodological solution.

In this study, we apply two comprehensive analyzes to them to determine the most efficient MCDM method: first, their relationship to real life, and second, their rank reversal (RR) performance. Accordingly, we present an objective comparison of nine (including classical, popular, and new) MCDM methods. Between the years 2019–2021, we measured the MCDM-based FP of 10 quarters (a separate performance was measured for each quarter) of 140 manufacturing companies from different sectors registered on the Stock Exchange (BIST). We evaluated the relationship between simultaneous price rankings and MCDM. Also, as a second criterion, we propose how to measure the RR performance of MCDM methods for a given problem with a unique robust procedure.

In conclusion, we propose two specific and objective evaluation procedures that confirm each other, showing which method is more successful and more capable for decision-makers. We did not base the validity of the analysis results on only one problem, the MCDM method or the weighting method. Unlike many previous studies, this comprehensive study uses 10 different real scenarios, nine different MCDMs, two weighting methods, and more than one hundred alternatives to test the reliability and validity of this procedure.

We evaluate that our study will contribute to the literature methodologically and practically by using two objective criteria simultaneously for MCDM comparison and selection, which is a challenging subject. In this study, first of all, BIST manufacturing sector will be calculated using various MCDM methods. In the next step, MCDM methods will be compared according to the correlations between financial performance results and stock returns. To gain a more robust insight, a separate comparison metric will be proposed based on the RR findings produced by the MCDM methods. It is thought that this capacity assessment to be made with these two criteria, which express the relationship and consistency of MCDM methods with real life, can fill an important gap in the literature. In this study, first of all, a comprehensive literature review will be made under various headings. Later, the research methodology will be explained in all its aspects. Findings will be presented next and they will be evaluated in the discussion. Finally, final evaluations will be made in the conclusions section.

## LITERATURE REVIEW

In this section, we will first examine some of the classical views in the literature on rationality on which the concepts of selection, comparison, and capacity for an appropriate MCDM are based. Second, we look at MCDM-based FP measurement and evaluation studies. Next, we will review some recent innovative studies that have addressed the relationship between FP and SR and evaluate this relationship as a capacity indicator for MCDMs. Finally, we review the literature on RR, an MCDM evaluation, and a benchmark.

## MCDM evaluation methodology

Among the 200+ MCDM methods (*Cinelli et al., 2022*), there are simple and primitive ones such as SAW (Simple Additive Weighting method), as well as algorithmically sophisticated methods such as PROMETHEE-2 (Preference Ranking Organization Method for Enrichment Evaluation), VIKOR (VIseKriterijumska Optimizacija I Kompromisno Resenje), or TOPSIS (Technique for Order of Preference by Similarity to Ideal Solution). However, while the MCDM method selection and MCDM evaluation methodology are mostly associated with the sophistication of input-based algorithms, sufficient data analytics have not been conducted regarding the outputs (outcomes) they produce. In this sense, it is not surprising that the existing literature emphasizes that the problem is difficult or has no definitive solution. (*Triantaphyllou, 2000*). On the other hand, there are many suggestions in the literature about what approach should be adopted when determining any MCDM method. For example, there are some studies claiming that SWOT (Weaknesses, Strengths, Threats and Opportunities) analysis of MCDM methods is vital. The advantages and disadvantages of each of these methods need to be determined. Then, creating new methods that can effectively combine strengths while eliminating weaknesses can increase efficiency (*Velasquez & Hester, 2013*). Here, strengths and weaknesses are sometimes relative, which can be another problem. On the other hand, comparative analyzes of the methods show that none of the MCDM methods are perfect. For this reason, it has been suggested to apply more than one method to the same problem in order to give a more comprehensive result to the decision makers and it is still widely applied (*Mulliner, Malys & Maliene, 2016*). Although it is unlikely to be validated, combining different methods as an approach can overcome the shortcomings of these methods. There are many problematic issues that complicate the MCDM Evaluation Methodology. The first problem is which of the assumptions based on outranking, value/benefit, and distance should be preferred. Selection of MCDM type, normalization type, weighting technique type, appropriate threshold value, and preference function type is uncertain problem. Purging a method from "rank reversal" and "compensatory" is another important problem. The lack of a standard procedure for sensitivity, robustness, and accuracy analysis can be counted as one of the final problems.

Struggles continue intensely to determine an objective and understandable framework procedure on how to choose the most appropriate method (*Guarini, Battisti & Chiovitti, 2018*). Some of the findings are remarkable and can be evaluated. Some "outranking methods" such as PRMOTHEE-2 are less compensatory (one or more criteria suppressing other criteria and distorting fairness), which is a positive feature. On the other hand, compared to its competitors, AHP (Analytic Hierarchy Process) can produce consistency, time and energy problems as it makes many pairwise comparisons with subjective expert opinion. TOPSIS is based on ideal values (positive and negative) and it may produce higher rank reversal problem. Many MCDM methods transform and distort information with normalization, which is the most important cause of rank reversal (*Wu & Abdul-Nour, 2020*). On the other hand, sensitivity analysis, which is claimed to measure robustness for the comparison problem of MCDMs, is proposed. (*Haddad, Sanders & Tewkesbury, 2020*).

Moreover, it is emphasized that axiomatics for the selection of MCDM methods can be a useful way to improve the decision-making process (*Leoneti, 2016*). According to some authors, the selection of MCDM can be an erroneous and problematic issue from time to time. Good guidance is essential for its selection. There is a solution point that all previous studies have pointed out insightfully, which is that the necessity of a concrete, clear, and objective evaluation framework is imperative for this problem (*Cinelli et al., 2022*).

On the other hand, the capacity of MCDM methods to represent real-life scenarios is becoming more important than before (*Munier, 2006*). Therefore, MCDM methods should be evaluated not only with their potential capacity but also with their capacity to capture real life. MCDM methods that establish a better and more consistent relationship with real-life rankings can be considered successful. Therefore, for MCDM methods, not only their methodological capabilities (calculation procedure) should be evaluated, but also their output-based (results) capabilities. Moreover, it can be said that the rank reversal performance of MCDM methods for a particular problem has not been adequately addressed in a comparative and comprehensive manner.

## Financial performance (FP) and MCDM

Multiple and sometimes contradictory purposes of financial criteria require using the consensus solution of the MCDM paradigm for computation. The idea of reducing FP to a single score with MCDM dates back more than 20 years in the literature.

*Ertuğrul & Karakaşoğlu (2009)* evaluated the FP of 15 cement companies based on 18 criteria and using fuzzy AHP and TOPSIS methods. *Wang (2014)* evaluated the FPs of three container shipping companies operating in the field of shipping in Taiwan using fuzzy multi-criteria decision making techniques. In his study, *Wang (2014)* first dealt with five periods with 21 ratios divided into clusters and then calculated representative indices from these clusters. Then, these representative indices were used as evaluation criteria, the FPs of the firms were evaluated using the fuzzy TOPSIS method, and then the firms were ranked according to their financial performance. *Rezaie et al. (2014)* evaluated the FPs of 27 listed companies operating in the cement sector between 2008–2009 using 13 ratios. In the study, companies are listed with the VIKOR method. On the other hand, *Pineda et al. (2018)* developed an integrated model for the performance evaluation of airline companies that uses data mining and MCDM methods together. They evaluated the performance of 12 air companies using 11 criteria divided into four main groups. They used a hybrid model, applying a combination of DRSA, DEMATEL, DANP, and VIKOR methods to rank and identify financial and operational critical factors. Similarly, *Yalçın & Ünlü (2018)* used the objective weighting method (CRITIC) and equal weight (MW) method together in their study to evaluate the initial public offering performance of companies and ranked the firm performances by evaluating them with the VIKOR method. In their studies, nine ratios were used for 16 companies, taking into account the period of 2010–2012. *Akgün & Soy Temür (2016)* made the 6-year FP evaluation of two airline companies in the BIST transportation sector between 2010 and 2015 using the TOPSIS method and then compared the findings. Similarly, *Metin, Yaman & Korkmaz (2017)* calculated the ratios of 11 energy companies in the BIST according to their financial statements between 2010 and

2015, and then evaluated the FPs of these companies using TOPSIS and MOORA methods. In their study, they concluded that the FPs of companies changed in both methods, but they observed that only three companies had the same performance ranking for both methods. Finally, *Özçelik & ve Küçükçakal (2019)* evaluated the FPs of seven financial leasing and factoring companies traded on the BIST for the period 2009–2016 using the TOPSIS method. In this analysis, they used six financial ratios and identified higher performing companies.

In MCDM-based FP studies, there is generally a lack of objective justification for choosing and using an MCDM method. Some other remarkable results that we can observe in FP measurement studies based on MCDM can be listed as follows (*De Almeida-Filho, De Lima Silva & Ferreira, 2020*): First, the number of studies using MCDM methods is constantly increasing from the past to present. Secondly, while TOPSIS was the most used MCDM method in performance studies, similarly, profitability and risk-based financial criteria were preferred more frequently. Third, FP in general is one of the most studied subjects in finance based on MCDM.

## The relationship between financial performance and stock return

In addition to the above-mentioned case studies in which FP was calculated integrally with MCDM methods, there are few other studies comparing the obtained FP results with price. SR or return on share (return on capital) is defined as the percentage change in price. For instance, *Sakarya & Aytekin (2013)* used 10 financial ratios to determine the relationship between FP and stock returns (SR) of deposit banks traded in BIST during the period between 2007 and 2011. In this sense, they used PROMETHEE method as one of the MCDM methods and Spearman rank correlation as statistical method. On the other hand, *Işık (2019)* used the entropy method to determine the weights of financial variables, and applied TOPSIS method to evaluate the performance of the companies in his study, in which he investigated the relationship between SR and the FP of the companies in the BIST 30 Index. He analyzed the relationship between financial performance and stocks by correlation analysis. *Baydaş & Pamučar (2022)*, *Baydaş, Elma & Pamučar (2022)* stated in his study that there is a consistent and continuous relationship between MCDM-based FP and stock return under certain constraints. In other words, some methods stably capture the relationship between two variables better. This shows that methods that capture real life have a special capacity.

## Rank reversal

RR is a phenomenon where the order (rank) of some alternatives in the new scenario is reversed when alternative(s) are added or removed. The reasonable suspect for the RR problem is usually normalization, but not just that (*Barzilai & Golany, 2017*). Since normalization distorts the original data in the first decision matrix, it violates the principle of independence from irrelevant alternatives (PIIA) (*Mufazzal & ve Muzakkir, 2018*). To give an example, in the first case, alternative A is higher in rank than alternative B in an MCDM ranking array. After adding or subtracting a different alternative to the ranking, alternative A falls behind alternative B in the ranking. Such a situation also means that the

MCDM method produces an RR problem. In the literature, the issue of RR, which is an important problem related to MCDM methods in many scientific fields, has been researched and proposed solutions (*Belton & ve Gear, 1983*; *Keshavarz-Ghorabaee et al., 2018*; *Bączkiewicz et al., 2021*; *Agrawal, 2021*). In general, rank reversal is a valid problem for MCDM methods to some extent.

Comparative determination of the RR degree (performance) for MCDM methods for a given problem is an important requirement. The classical approach in this regard consists of observing the results by adding or subtracting an alternative. However, it is also possible to approach the subject more comprehensively and statistically. In their study, *Mufazzal & ve Muzakkir (2018)* recommended using the Spearman rank correlation, which is a statistical method, to measure RR sensitivity. In this study, the RR degree of MCDM methods was measured objectively with Spearman correlation. The procedure used here is unique. Any MCDM calculations are made first with the available alternatives. Then, as many alternatives as possible are discarded from the system, and MCDM is calculated again with the remaining ones. If the correlation result of the common alternatives in two different MCDM series is one (1), it is said that there is no RR problem. The closer to zero, the greater the RR problem.

For example, in this study, half of the 140 alternatives are discarded from the system. The correlation between the two 70 alternatives gives the degree of RR. Note that we are dealing with subtracting alternatives for the first sequence and adding alternatives for the second sequence. Also, the purpose of adding and removing a large number of alternatives here is to stress test an MCDM method to force it to RR error (production) or to measure its RR immunity. This is a procedure that has been proposed and successfully applied for the first time in the literature. Moreover, it will be possible to compare MCDM methods with this objective RR measurement framework.

## The literature gaps and suggestions

Our study has potential to fill some gaps in the literature by a more comprehensive analysis procedure. Our suggestions and solutions for these gaps are listed as follows:

- It is a difficult and chronic problem to determine which of the MCDM methods is more suitable and which one should be chosen. However, in this study, we made an application with an indirect but practical, undeniable and objective criterion. Accordingly, we preferred only one MCDM method in FP calculation that produces a better statistical correlation with price. For example, some methods provide a low level and some methods provide a high-level relationship between two variables, which we obviously know that they have significant relationship. The different findings of MCDM methods show their success, thus, we can compare different MCDM methods according to their capacity to capture real-life, and then we can choose the most appropriate method.

- In this study, we will try to explore for the first time the RR sensitivity of MCDM methods for a particular problem. We will search for valuable clues with MCDM analytics.

In the few previous studies using this procedure, the success of certain MCDM methods stands out (*Baydaş & Pamučar, 2022*). Therefore, this study is aimed to see the big picture better and to reveal the capabilities of different methods. In addition to the methods used in previous studies, some new (*e.g.*, COCOSO and CRADIS), classical (*e.g.*, TOPSIS, VIKOR), and popular (*e.g.*, FUCA and MABAC) methods were used in this study. When a decision maker wants to compare MCDM methods comprehensively, it is undoubtedly logical to use classical, popular, new, as well as all schools ("outranking", utility, value, distance-based, *etc.*) in the literature. Accordingly, we used representatives of all MCDM types in this study, including the new methods. There are more than 200 MCDM methods and the sample of MCDM methods in this study has a very high ability to represent the universe.

# RESERCH METHODOLOGY

In this study, in which new objective criteria are proposed for the methodology for evaluating MCDM methods, which is a rather complicated subject, MCDM analytics has been conducted in order to obtain valuable useful information from the data sets. In this study, we first measured the financial performance (FP) of 140 manufacturing companies traded in the BIST using MCDM methods. FP was measured with nine MCDM methods, which are COCOSO, CRADIS, CODAS, COPRAS, ELECTRE-3, FUCA, MABAC, TOPSIS, and VIKOR. Since there is no complete agreement on which one should be used in the calculation of VIKOR, we added the S-R and Q derivatives/components separately to the results. We have seen that its derivatives have different capacities. SWARA, a subjective method, and CRITIC, an objective method, were used separately for criterion weighting. Here, as readers can guess, we sought a valid answer to the question of whether objective dynamic or subjective static methods are more successful when determining the weight coefficient as a separate main purpose. Next, the Spearman rank correlation analysis was applied to reveal which MCDM method establishes the current statistical relationship between price (stock return/capital gain) and FP. As a second MCDM evaluation method, we followed an innovative procedure and objectively proved the rank reversal (RR) performance of nine MCDM methods with findings in 10 different real scenarios.

## Performance metrics

Information about the FP measurement criteria and their calculation details are explained below and shown in Table 1. The ultimate goal of the firms in the stock market is to maximize the share price and hence the firm value. In general, performance indicators that can establish a close relationship with the share price in every period are not many and they are generally some basic and vital metrics based on risk, value generation, and profitability. Moreover, they are not static but change-based (*Baydaş & Pamučar, 2022*).

### Altman-Z score

The original use of the Altman-Z score was actually to determine and evaluate the degree of closeness of a firm to bankruptcy, which has been used frequently for more than half a century. As used in this study, its increase or decrease (change) value generally includes a

**Table 1 Calculation details of the criteria used in the study.**

| Ratio | Calculation | Formula no. | References |
|---|---|---|---|
| MV/BV | Marketing Value/Equity | (1) | *Stewart (2013)* |
| ROE | Net Profit/Equity Value | (2) | *Bodie, Kane & Marcus (2018)* |
| MVA margin | (MVt - Inv. Cap. t-1)/Sales t-1 | (3) | *Stewart (2013)* |
| ALTMAN-Z score | 1.2A + 1.4B + 3.3C + 0.6D + 1.0E<br>A = Working Capital/Total Assets<br>B = Retained Profits/Total Assets<br>C = Earnings Before Interest and Tax/Total Assets<br>D = Market Value of Assets/Total Liabilities<br>E = Sales/Total Assets | (4) | *Carton (2004)* |
| *Stock return* | (Current Stock Price - Previous Period Base Price)/Base Price (5) | | *Carton & Hofer (2006)* |

linear and significant relationship with SR (*Carton, 2004*). The Altman-Z score is actually a very important risk indicator that can also measure financial success. As it is a benefit-oriented criterion, higher values are always expected for this indicator.

### ROE (net profit/equity)

ROE, defined as the ratio of net profit to equity, is one of the most classic and successful metrics for a firm that focuses on the efficiency of profits *per se*. It is frequently used and recommended to evaluate FP (*Bodie, Kane & Marcus, 2018*). Its change value is usually in a linear and meaningful relationship with price.

### MV/BV (market value/book value)

It can be expressed as the ratio of market value to equity. It expresses the equivalent of the accounting book value as the market value (*Stewart, 2013*). This is a result of pricing equity in the market. High market pricing of book value by investors means there are high positive earnings expectations for that stock.

### MVA margin

MVA margin is the ratio of market value added (MVA) to net sales. MVA is a classic value-based measure for assessing FP. MVA Margin is a benefit-oriented criterion derived from MVA (*Stewart, 2013*). In other words, the MVA margin measures the value created or eroded by net sales. It gives an important insight into the company in general, whether sales translate into value.

## MCDM methods

In this study, nine MCDM methods were calculated for 140 companies and 10 separate quarters. First of all, financial performance results based on MCDM were evaluated to show the relationship with stock returns. Then, for the rank reversal performance calculation, in addition to the current MCDM results for the 140 companies at hand, the financial performance was calculated again with MCDM methods for 70 companies. Spearman correlation results between 70 common companies in the two lists were compared for the RR level. Therefore, since a total of 180 MCDM calculations were made,

it was not possible to show their results in the article or in the appendices (because it took up too many pages). Brief descriptive explanations of MCDM methods are given below. Due to a large number of methods, the steps related to the calculation steps of the formulas are presented as tables in the Supplemental Files. Brief descriptive explanations of MCDM methods are given below. Due to a large number of methods, the steps related to the calculation steps of the formulas are presented as tables in the Supplemental Files. Table 2 shows the names, descriptions and references of the MCDM and weighting methods used in this study.

## Weighting method

In order to understand which one is more efficient as a weighting method, both objective and subjective methods were preferred in this study. These are SWARA (Step-Wise Weight Assessment Ratio Analysis) and CRITIC (Criteria Importance Through Inter Criteria Correlation) whose information is given in the table above. SWARA is a relatively new criterion weighting method compared to the classical others, which is a widely used weighting technique. It is also a subjective method as it recommends getting an expert opinion. SWARA was preferred in this study because it better reflects the expert opinion and provides a broad perspective. Basically, value-based ratios should have a higher weighting coefficient than accounting-based ratios. SWARA is a method that has the capacity to consider this situation. In addition to SWARA, CRITIC weighting methods were also calculated in our study and all final results were compared to gain further insight.

## FINDINGS AND DISCUSSIONS

### Data set and experimental design

In order to measure financial performance with MCDM methods, 140 manufacturing companies operating in the manufacturing sector in the Turkish BIST were taken into account and four different performance indicators belonging to these companies were determined as decision criteria. The performances of these companies in 10 quarters (2019–2021) were taken as the subject of the study. MCDM-based financial performance (FP) of companies (for each quarter) was calculated separately. The financial data of the companies were taken through FİNNET Commercial data software. COCOSO, CRADIS, CODAS, COPRAS, ELECTRE-3, FUCA, MABAC, TOPSIS and VIKOR (S,R,Q) methods were used as MCDM methods to measure the performance of the companies. Then the relationship of MCDM results with SR (stock return) was obtained by Spearman rank correlation coefficient for each base period.

The steps of the experimental process of the study are explained below and summarized in Fig. 1.

### *First stage*

It includes the steps consisting of the determination of the criteria, the creation of the first decision matrix, which is the basis for the calculations, data normalization and MCDM calculations.

**Table 2 Methods used in this study.**

| MCDM methods | Explanation | References |
|---|---|---|
| COCOSO | The proposed approach is based on an integrated simple additive weighting and exponentially weighted product model. Recommended as a summary of reconciliation solutions. | *Yazdani et al. (2018)*, *Badi, Jibril & Bakır (2022)*, *Torkayesh et al. (2021)* |
| CRADIS | The philosophy of the CRADIS method is to determine the deviations of the alternatives from the ideal and anti-ideal solution. It can be said that this method is a combination of steps drawn from ARAS, MARCOS and TOPSIS methods. | *Puška, Stević & Pamučar (2022)*, *Puška & Stojanović (2022)* |
| CODAS | Since the overall performance of an alternative in CODAS is measured by its distance from the negative ideal point (NIS), each pair of alternatives is compared. In this method, the first criterion used in the superiority of alternatives to each other is the Euclidean distance and taxi distance of the alternatives considered from the negative ideal. | *Wang et al. (2020)* |
| COPRAS | In this method, how useful or unhelpful an alternative is compared to other alternatives is defined as a percentage. The higher the priority of the analyzed alternative, the higher the degree of utility. Thus, the degree of utility is determined by comparing each alternative with the most efficient alternative. | *Zavadskas & Kaklauskas (1996)*, *Wang et al. (2020)*, *Tripathi et al. (2022)*, *Ecer (2021)* |
| FUCA | This method is based on ranking the alternatives for each criterion. The first row has the best value, while the last row (n) is assigned the worst value. Then, the weighted sum of the values for each solution point is calculated and the solution with the smallest total value is the best-chosen solution. | *Mendoza et al. (2011)*, *Wang & Rangaiah (2017)*, *Baydaş & Pamučar (2022)*, *Do (2022)* |
| ELECTREIII | Using the satisfaction and dissatisfaction criteria at the same time, ELECTRE-3 also has both incompatibility and compatibility indices. Therefore, it can be considered as a method of listing the bilateral relations between alternatives. | *Wang & Rangaiah (2017)* |
| TOPSIS | In the TOPSIS method developed by Hwang and Yoon, the best alternative chosen is also the closest to the positive ideal solution and the farthest to the negative ideal solution. This method, based on vector distance and ideal values, is the most widely adopted classical method in various fields and in the nearly half-century history of MCDM. | *Çınar (2004)*, *De Almeida-Filho, De Lima Silva & Ferreira (2020)* |
| MABAC | The descriptive definition for MABAC, one of the most popular methods in the last decade, is the distance of each non-dominant solution from the target's boundary zoom area. | *Pamučar & Ćirović (2015)*, *Wang et al. (2020)* |
| VIKOR | A popular and sophisticated method, VIKOR is based on consensus and ranking assumptions. For this purpose, a multi-criteria ranking index is created over the alternatives and their closeness to the ideal solution is calculated and compared. It is a recommended method in situations where compromise solutions must be used to solve problems effectively and efficiently, especially in scenarios where conflict and incompatibility occur. | *Wang & Rangaiah (2017)* |
| Weighting Method | | |
| SWARA | It is based on ranking the criteria in decreasing order of importance by the decision maker (the first criterion is the most important, the last criterion is the least important). | *Keršulienė, Zavadskas & Turskis (2010)*, *Wang et al. (2020)*, *Stević et al. (2022b)* |
| CRITIC | CRITIC is an objective method that takes into account the standard deviation of the criteria and the dependence (correlation) between them. | *Diakoulaki, Mavrotas & Papayannakis (1995)* |

## Second stage

In the second stage, the final results of the MCDM methods are evaluated. MCDM methods are compared in terms of their relation to SR and RR performance. Spearman

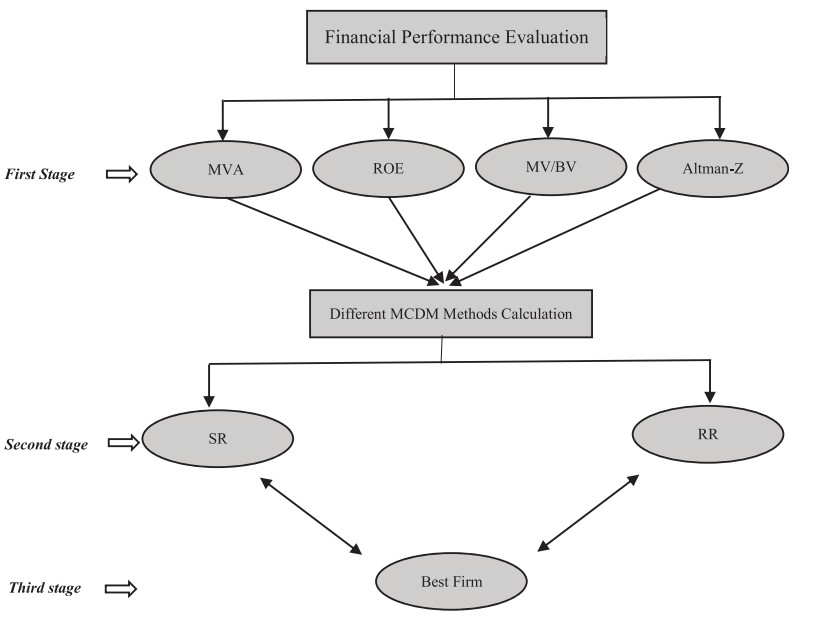

**Figure 1  Flowchart of the study.**               

correlation analysis is used for both of these criteria. In addition, two different weighting methods are compared.

### Third stage

In the third stage, the evaluation methodology is applied meticulously. Ultimately, the best companies are determined for each quarter. For this, first, an MCDM must be selected from among the alternatives. For MCDM selection, methods that show joint success in both criteria are examined. In addition, it is determined which of the weighting methods will be decided. Thus, based on the appropriate method, the best company is selected and recommended to the decision maker. A stock selection recommendation is beyond our scope here because our purpose is purely methodological. But users can also use the methodology here as a partial dimension of fundamental analysis.

## Findings

### Determination of weighting coefficient for MCDM based FP criteria

In fact, although the classical static ratios give an idea, a general evaluation of the change ratios of the overall financial performance of the companies and a stock-focused FP is a good guide for us. In order to make a fair comparison, common criterion weighting coefficients for MCDM methods should be obtained. For this reason, SWARA, which is one of the subjective techniques, and the CRITIC method, which is an objective method, were used in this study, which was made with the contributions of three experts. According to the analysis findings, SWARA was adopted in all assessments, as the SWARA method (as will be proved later) is obviously and dominantly better than the CRITIC method. Calculation results for the SWARA criterion weighting coefficient are as follows: Altman-Z Score 0.163; ROE 0.163; MV/BV is 0.245, MVA Margin is 0.428. CRITIC weighting coefficients, which is an objective method, are shown in Table 3 below. In fact, another

**Table 3  CRITIC criteria weighting results.**

|  | Q1 | Q2 | Q3 | Q4 | Q5 | Q6 | Q7 | Q8 | Q9 | Q10 |
|---|---|---|---|---|---|---|---|---|---|---|
|  | 2019/03 | 2019/06 | 2019/09 | 2019/12 | 2020/03 | 2020/06 | 2020/09 | 2020/12 | 2021/03 | 2021/06 |
| ALTMAN-Z score | 0.194 | 0.176 | 0.171 | 0.245 | 0.231 | 0.183 | 0.173 | 0.205 | 0.210 | 0.156 |
| ROE | 0.306 | 0.325 | 0.354 | 0.252 | 0.190 | 0.302 | 0.343 | 0.375 | 0.286 | 0.461 |
| MV/BV | 0.333 | 0.249 | 0.300 | 0.226 | 0.182 | 0.330 | 0.270 | 0.249 | 0.200 | 0.187 |
| MVA margin | 0.167 | 0.249 | 0.179 | 0.276 | 0.397 | 0.185 | 0.213 | 0.171 | 0.304 | 0.196 |

**Table 4  Rho coefficients showing the relationship between SR rankings and SWARA integrated MCDM rankings.**

|  | Q1 | Q2 | Q3 | Q4 | Q5 | Q6 | Q7 | Q8 | Q9 | Q10 |  |
|---|---|---|---|---|---|---|---|---|---|---|---|
|  | 2019/03 | 2019/06 | 2019/09 | 2019/12 | 2020/03 | 2020/06 | 2020/09 | 2020/12 | 2021/03 | 2021/06 | Mean |
| FUCA | 0.503 | 0.479 | 0.487 | 0.697 | 0.673 | 0.717 | 0.852 | 0.861 | 0.58 | 0.672 | 0.652 |
| Q (VIKOR) | 0.432 | 0.392 | 0.449 | 0.525 | 0.475 | 0.6 | 0.697 | 0.794 | 0.351 | 0.571 | 0.528 |
| TOPSIS | 0.395 | 0.391 | 0.367 | 0.382 | 0.592 | 0.471 | 0.613 | 0.846 | 0.564 | 0.646 | 0.526 |
| COCOSO | 0.374 | 0.421 | 0.309 | 0.503 | 0.484 | 0.599 | 0.671 | 0.611 | 0.361 | 0.638 | 0.497 |
| MABAC | 0.366 | 0.421 | 0.294 | 0.469 | 0.488 | 0.54 | 0.642 | 0.724 | 0.361 | 0.638 | 0.494 |
| S (VIKOR) | 0.366 | 0.421 | 0.294 | 0.469 | 0.488 | 0.54 | 0.642 | 0.724 | 0.361 | 0.638 | 0.494 |
| CRADIS | 0.366 | 0.421 | 0.294 | 0.469 | 0.488 | 0.54 | 0.642 | 0.724 | 0.361 | 0.638 | 0.494 |
| R (VIKOR) | 0.299 | 0.227 | 0.458 | 0.56 | 0.367 | 0.554 | 0.667 | 0.698 | 0.25 | 0.42 | 0.45 |
| CODAS | 0.2 | 0.435 | 0.409 | 0.192 | 0.572 | 0.056 | 0.512 | 0.853 | 0.487 | 0.495 | 0.421 |
| COPRAS | 0.3 | 0.099 | 0.011 | 0.281 | 0.094 | 0.089 | 0.682 | 0.618 | 0.81 | 0.527 | 0.351 |
| ELC3 | 0.123 | 0.357 | 0.241 | 0.318 | 0.308 | 0.339 | 0.324 | 0.607 | 0.087 | 0.541 | 0.324 |

invisible main purpose of this study is to make a comparison for objective and subjective methods.

It should be noted here that the SWARA method is static/single for all periods and the CRITIC method is dynamic/variable.

***Comparison of the calculated MCDM methods and analysis findings of the suggested evaluation methodology***

Recently, some researchers, who have been making MCDM calculations with financial-based data, are interested in finding the most appropriate MDCM method by using the relationship between FP and SR as a tool. It may be possible for an MCDM method that better models real life to better capture SR rankings. At this point, the statistical relationship between FP and SR rankings calculated by Spearman's rank correlation method can be evaluated. For comparison, nine new, classical and popular MCDM methods were compared statistically with SR separately. The final Spearman rho correlation coefficient results obtained with this approach are shown in Tables 4 and 5 below.

FUCA is able to produce a better relationship to real life compared to other methods and has more consistent success in most quarters and averages. Table 6 below confirms

**Table 5 Rho performance ranking of SWARA-based MCDM results.**

|         | Q1 | Q2 | Q3 | Q4 | Q5 | Q6 | Q7 | Q8 | Q9 | Q10 | Rank mean |
|---------|----|----|----|----|----|----|----|----|----|-----|-----------|
| FUCA    | 1  | 1  | 1  | 1  | 1  | 1  | 1  | 1  | 2  | 1   | 1.1       |
| Q       | 2  | 7  | 3  | 3  | 8  | 2  | 2  | 4  | 9  | 7   | 4.7       |
| COCOSO  | 4  | 3  | 6  | 4  | 7  | 3  | 4  | 10 | 5  | 3   | 4.9       |
| CRADIS  | 5  | 4  | 7  | 5  | 4  | 5  | 6  | 5  | 6  | 4   | 5.1       |
| TOPSIS  | 3  | 8  | 5  | 8  | 2  | 8  | 9  | 3  | 3  | 2   | 5.1       |
| MABAC   | 6  | 5  | 8  | 6  | 5  | 6  | 7  | 6  | 7  | 5   | 6.1       |
| CODAS   | 10 | 2  | 4  | 11 | 3  | 11 | 10 | 2  | 4  | 10  | 6.7       |
| R       | 9  | 10 | 2  | 2  | 9  | 4  | 5  | 8  | 10 | 11  | 7         |
| S       | 7  | 6  | 9  | 7  | 6  | 7  | 8  | 7  | 8  | 6   | 7.1       |
| COPRAS  | 8  | 11 | 11 | 10 | 11 | 10 | 3  | 9  | 1  | 9   | 8.3       |
| ELC3    | 11 | 9  | 10 | 9  | 10 | 9  | 11 | 11 | 11 | 8   | 9.9       |

**Table 6 Rho performance ranking of SWARA-based MCDM results.**

|          | Q1    | Q2    | Q3    | Q4    | Q5    | Q6    | Q7    | Q8    | Q9    | Q10   | Mean  |
|----------|-------|-------|-------|-------|-------|-------|-------|-------|-------|-------|-------|
| FUCA     | 0.994 | 0.998 | 0.997 | 0.999 | 0.994 | 0.984 | 0.996 | 0.998 | 0.997 | 0.998 | 0.995 |
| CRADIS   | 0.995 | 0.997 | 1     | 0.983 | 0.969 | 0.987 | 0.987 | 0.976 | 0.942 | 0.976 | 0.981 |
| MABAC    | 0.995 | 0.997 | 1     | 0.983 | 0.969 | 0.987 | 0.987 | 0.976 | 0.942 | 0.976 | 0.981 |
| S        | 0.995 | 0.997 | 1     | 0.983 | 0.969 | 0.987 | 0.987 | 0.976 | 0.942 | 0.976 | 0.981 |
| COCOSO   | 0.995 | 0.998 | 1     | 0.967 | 0.987 | 0.959 | 0.969 | 0.977 | 0.92  | 0.976 | 0.974 |
| TOPSIS   | 0.938 | 0.978 | 0.976 | 0.968 | 0.99  | 0.958 | 0.911 | 0.923 | 0.98  | 0.994 | 0.961 |
| Q        | 0.906 | 0.996 | 1     | 0.971 | 0.982 | 0.944 | 0.901 | 0.867 | 0.992 | 0.959 | 0.951 |
| CODAS    | 0.993 | 0.982 | 0.999 | 0.999 | 0.857 | 0.994 | 0.974 | 0.875 | 0.84  | 0.994 | 0.950 |
| ELECTRE3 | 0.966 | 0.985 | 0.995 | 0.929 | 0.91  | 0.947 | 0.915 | 0.877 | 0.857 | 0.901 | 0.928 |
| R        | 0.495 | 1     | 1     | 1     | 1     | 1     | 0.647 | 0.639 | 1     | 0.906 | 0.868 |
| COPRAS   | 0.267 | 0.097 | 0.902 | 0.956 | 0.984 | 0.998 | 0.471 | 0.934 | 0.867 | 0.909 | 0.738 |

that FUCA is the winner in all the remaining periods except for one period, according to the rho results. While VIKOR's Q ranking results are very good, R and S results are not so good. When these three results are averaged, VIKOR is not in the top three. The top three successful methods in terms of sequencing performance are FUCA, CoCoSo, and CRADIS methods. It is seen that ELECTRE-3 and COPRAS are in last place. The fact that COPRAS is first in the ninth quarter and third in the seventh quarter, but generally second to last, shows how volatile this method is. We consider that the results will be more positive when an appropriate normalization selection and some minor computational improvements are made to this method. We have similar thoughts about CODAS. Despite the volatile nature of financial data, the very stable and consistent performance of the MABAC method is admirable and remarkable. Although TOPSIS is the most widely adopted and popular method in MCDM history, its performance is mediocre.

**Table 7 Ranking of SWARA-based MCDM methods by rank reversal performance.**

|          | Q1 | Q2 | Q3 | Q4 | Q5 | Q6 | Q7 | Q8 | Q9 | Q10 | RR mean |
|----------|----|----|----|----|----|----|----|----|----|-----|---------|
| FUCA     | 5  | 3  | 8  | 3  | 2  | 7  | 1  | 1  | 2  | 1   | 3.3     |
| CRADIS   | 2  | 4  | 2  | 4  | 7  | 4  | 2  | 3  | 5  | 5   | 3.8     |
| COCOSO   | 1  | 2  | 1  | 9  | 4  | 8  | 6  | 2  | 8  | 4   | 4.5     |
| MABAC    | 3  | 5  | 3  | 5  | 8  | 5  | 3  | 4  | 6  | 6   | 4.8     |
| R        | 10 | 1  | 5  | 1  | 1  | 1  | 10 | 11 | 1  | 10  | 5.1     |
| S        | 4  | 6  | 6  | 6  | 9  | 6  | 4  | 5  | 7  | 7   | 6       |
| CODAS    | 6  | 9  | 7  | 2  | 11 | 3  | 5  | 9  | 11 | 2   | 6.5     |
| TOPSIS   | 8  | 10 | 10 | 8  | 3  | 9  | 8  | 7  | 4  | 3   | 7       |
| Q        | 9  | 7  | 4  | 7  | 6  | 11 | 9  | 10 | 3  | 8   | 7.4     |
| COPRAS   | 11 | 11 | 11 | 10 | 5  | 2  | 11 | 6  | 9  | 9   | 8.5     |
| ELECTRE3 | 7  | 8  | 9  | 11 | 10 | 10 | 7  | 8  | 10 | 11  | 9.1     |

It is clear how variable the quarterly ranking results are for methods other than FUCA and MABAC. It is a matter of curiosity whether the successful performance of FUCA and the new methods CoCoSo and CRADIS will also be valid for the other evaluation criteria, RR. Table 7 below shows the RR sensitivity analysis results of MCDM methods in each quarter and the general mean. In this study, a very innovative and practical procedure was determined. We suggest a more objective assessment than classical RR measures, as we know that RR sensitivity is a serious consistency issue that can distort rankings. In classical measures, an alternative is added and subtracted, and then it is observed by checking the individual alternatives whether the consistency in the ranking is impaired. However, such a measurement does not allow us to make a general conclusion about an MCDM method, so the final results are somewhat random. Moreover, individual control is a weak evaluation from a statistical point of view. In this study, we performed the RR sensitivity multiple times (10 times) for 10 quartiles. Moreover, we operated this for nine different methods. Moreover, we ultimately did not observe the alternatives and instead performed the Spearman correlation analysis. We continue to explain the procedure through our example. RR can be measured logically and objectively through Spearman rank correlation (*Mufazzal & ve Muzakkir, 2018*). We already have the final ranking results of all MCDM methods for 140 companies from previous analyses. Then, in order to calculate the RR, we divided 140 firms in the middle (we divide them in alphabetical order to be fair) and again made the MCDM calculations using the same decision matrix and weight coefficients of the first 70 firms. Then, we compared the MCDM results of 70 companies in the first list with the 70 companies in the second list with Spearman correlation analysis. Here, the correlation results will produce values between 0–1. If the result is close to 1 (one), it means that there is no RR problem at all. If the result is close to 0 (zero), it means that the RR problem increases. Notice that we subtract alternatives according to the first list and add alternatives according to the second list. Adding and removing as many alternatives as possible will produce more reliable and valid results for an MCDM method as it is almost a stress test. Users can also run a separate RR test for the second list, which we recommend.

**Table 8 Ranking of SWARA-based MCDM methods by rank reversal performance.**

|           | Q1    | Q2    | Q3    | Q4    | Q5    | Q6    | Q7    | Q8    | Q9    | Q10   | Rho mean |
|-----------|-------|-------|-------|-------|-------|-------|-------|-------|-------|-------|----------|
| COCOSO    | 0.278 | 0.362 | 0.266 | 0.423 | 0.435 | 0.458 | 0.46  | 0.414 | 0.232 | 0.521 | 0.384    |
| CODAS     | 0.134 | 0.453 | 0.309 | 0.157 | 0.416 | 0.03  | 0.214 | 0.851 | 0.357 | 0.535 | 0.345    |
| COPRAS    | 0.369 | 0.02  | 0.107 | 0.215 | 0.02  | 0.131 | 0.481 | 0.752 | 0.795 | 0.228 | 0.311    |
| CRADIS    | 0.271 | 0.369 | 0.237 | 0.379 | 0.432 | 0.375 | 0.413 | 0.473 | 0.179 | 0.526 | 0.365    |
| ELECTRE3  | 0.134 | 0.282 | 0.207 | 0.269 | 0.275 | 0.274 | 0.215 | 0.41  | 0.025 | 0.454 | 0.254    |
| FUCA      | 0.455 | 0.505 | 0.406 | 0.64  | 0.599 | 0.561 | 0.78  | 0.744 | 0.501 | 0.653 | 0.584    |
| MABAC     | 0.271 | 0.369 | 0.237 | 0.379 | 0.432 | 0.375 | 0.413 | 0.473 | 0.179 | 0.526 | 0.365    |
| Q         | 0.588 | 0.284 | 0.373 | 0.457 | 0.463 | 0.597 | 0.724 | 0.237 | 0.179 | 0.43  | 0.433    |
| R         | 0.739 | 0.215 | 0.388 | 0.54  | 0.366 | 0.635 | 0.783 | 0.028 | 0.254 | 0.3   | 0.424    |
| S         | 0.271 | 0.369 | 0.237 | 0.379 | 0.432 | 0.375 | 0.413 | 0.473 | 0.179 | 0.526 | 0.365    |
| TOPSIS    | 0.218 | 0.306 | 0.228 | 0.286 | 0.436 | 0.199 | 0.242 | 0.552 | 0.287 | 0.452 | 0.320    |

According to the results of Table 7 above, the method with the best performance according to RR, another MCDM evaluation criterion, is FUCA (as before). The RR analysis findings show that, in general, MCDM methods produce very low RR problems, the results are close to each other, but the presence of RR is present. It is noteworthy that the CRADIS, MABAC and S methods have the same degree of RR on average. Rarely in some quarters, some MCDM methods produced no RR at all. For VIKOR, while the Q method was prominent in the previous analysis, the S order was more successful in this analysis. There are six different quarters where the R method produces no RRs, but the others have low average performance because they produce high RRs. In parallel with the previous analysis, we observed that the performance of the derivatives of the VIKOR method is volatile here as well. We continue to think that minor computational improvements (revisions) are critical for VIKOR, which has a high potential. It is clear that TOPSIS, the most popular method, performed mediocrely in this analysis as well as before. COPRAS clearly stands out as the MCDM method that is exposed to the highest degree of RR problems. In both analyses, the performance of the CODAS, COPRAS, and ELECTRE-3 methods is low. If it is accepted that the RR problem is caused by normalization, it is recommended to use different normalization types for these methods.

According to the MCDM RR ranking performance findings in Table 8 above, the performance of the new methods CoCoSo and CRADIS, as well as FUCA, are the best. In the previous analysis, CRADIS was in the top three and continued its success here as well. When we make an evaluation based on the two criteria, the most successful methods are CRADIS and FUCA. It is not possible for these objective results, which we evaluated with two criteria for 10 different real-life scenarios, to be coincidental. It has also caught the reader's attention that we have run all analyzes based on the SWARA weighting method, which is a subjective method from the beginning. In fact, an objective method, CRITIC, was also used as an alternative in this study. But SWARA is obviously more successful when it comes to its relation to real life. This can be clearly seen in the table below. These findings show that it is critical to consult experts in the field to determine the weight

**Table 9 Comparison of CRITIC and SWARA methods.**

|  | Q1 | Q2 | Q3 | Q4 | Q5 | Q6 | Q7 | Q8 | Q9 | Q10 | Rho mean |
|---|---|---|---|---|---|---|---|---|---|---|---|
| FUCA-SW | 0.503 | 0.479 | 0.487 | 0.697 | 0.673 | 0.717 | 0.852 | 0.861 | 0.58 | 0.672 | 0.652 |
| FUCA-C | 0.455 | 0.505 | 0.406 | 0.64 | 0.599 | 0.561 | 0.78 | 0.744 | 0.501 | 0.653 | 0.584 |
| Q-SW | 0.432 | 0.392 | 0.449 | 0.525 | 0.475 | 0.6 | 0.697 | 0.794 | 0.351 | 0.571 | 0.528 |
| TOPSIS-SW | 0.395 | 0.391 | 0.367 | 0.382 | 0.592 | 0.471 | 0.613 | 0.846 | 0.564 | 0.646 | 0.526 |
| COCOSO-SW | 0.374 | 0.421 | 0.309 | 0.503 | 0.484 | 0.599 | 0.671 | 0.611 | 0.361 | 0.638 | 0.497 |
| CRADIS-SW | 0.366 | 0.421 | 0.294 | 0.469 | 0.488 | 0.54 | 0.642 | 0.724 | 0.361 | 0.638 | 0.494 |
| MABAC-SW | 0.366 | 0.421 | 0.294 | 0.469 | 0.488 | 0.54 | 0.642 | 0.724 | 0.361 | 0.638 | 0.494 |
| S-SW | 0.366 | 0.421 | 0.294 | 0.469 | 0.488 | 0.54 | 0.642 | 0.724 | 0.361 | 0.638 | 0.494 |
| R-SW | 0.299 | 0.227 | 0.458 | 0.56 | 0.367 | 0.554 | 0.667 | 0.698 | 0.25 | 0.42 | 0.45 |
| Q-CR | 0.588 | 0.284 | 0.373 | 0.457 | 0.463 | 0.597 | 0.724 | 0.237 | 0.179 | 0.43 | 0.433 |
| R-CR | 0.739 | 0.215 | 0.388 | 0.54 | 0.366 | 0.635 | 0.783 | 0.028 | 0.254 | 0.3 | 0.424 |
| CODAS-SW | 0.2 | 0.435 | 0.409 | 0.192 | 0.572 | 0.056 | 0.512 | 0.853 | 0.487 | 0.495 | 0.421 |
| COCOSO-CR | 0.278 | 0.362 | 0.266 | 0.423 | 0.435 | 0.458 | 0.46 | 0.414 | 0.232 | 0.521 | 0.384 |
| CRADIS-CR | 0.271 | 0.369 | 0.237 | 0.379 | 0.432 | 0.375 | 0.413 | 0.473 | 0.179 | 0.526 | 0.365 |
| MABAC-CR | 0.271 | 0.369 | 0.237 | 0.379 | 0.432 | 0.375 | 0.413 | 0.473 | 0.179 | 0.526 | 0.365 |
| S-CR | 0.271 | 0.369 | 0.237 | 0.379 | 0.432 | 0.375 | 0.413 | 0.473 | 0.179 | 0.526 | 0.365 |
| COPRAS-SW | 0.3 | 0.099 | 0.011 | 0.281 | 0.094 | 0.089 | 0.682 | 0.618 | 0.81 | 0.527 | 0.351 |
| CODAS-CR | 0.134 | 0.453 | 0.309 | 0.157 | 0.416 | 0.03 | 0.214 | 0.851 | 0.357 | 0.535 | 0.345 |
| ELC3-SW | 0.123 | 0.357 | 0.241 | 0.318 | 0.308 | 0.339 | 0.324 | 0.607 | 0.087 | 0.541 | 0.324 |
| TOPSIS-CR | 0.218 | 0.306 | 0.228 | 0.286 | 0.436 | 0.199 | 0.242 | 0.552 | 0.287 | 0.452 | 0.320 |
| COPRAS-CR | 0.369 | 0.02 | 0.107 | 0.215 | 0.02 | 0.131 | 0.481 | 0.752 | 0.795 | 0.228 | 0.311 |
| ELECTRE3-CR | 0.134 | 0.282 | 0.207 | 0.269 | 0.275 | 0.274 | 0.215 | 0.41 | 0.025 | 0.454 | 0.254 |

coefficient. Table 9 below shows the Relationship between SR Rankings and CRITIC-based MCDM Rankings.

It can be better seen in the Table 9 below that the CRITIC method weakens its relationship with the SR for any MCDM method. It is quite remarkable here that FUCA is less affected by the weight coefficient. The SWARA method is more successful when we take the period averages as a basis.

## Discussions

As is known, the methodology for evaluating MCDM methods is one of the most difficult and intricate issues. Because it is still not clear which MCDM method is superior and on what basis it was chosen. Even small objective steps to be taken in this regard are of great value. Original findings about the characteristics of MCDM methods will be instructive. Some previous recent authors (*Baydaş & Pamučar, 2022*) suggested that a stock return could be used as a reference for financial data. In this study, we add another one to this criterion, which is Rank reversal. Rank reversal is actually a previously well-known concept, but in no study has it been calculated as reliably and comprehensively as the statistical procedure here. In this study, we can say that we have taken a comprehensive

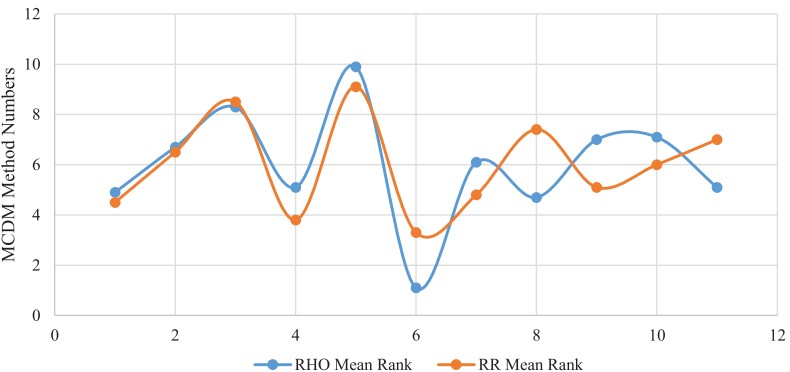

**Figure 2 Similarity of MCDM benchmarking tools.** Relationship with SR (rho) and rank reversal (RR).

inventory of rank reversal performance for MCDM methods (for financial data). With data analytics, large-scale data has been transformed into very useful information for the user. On the other hand, the fact that both criteria produce similar results at the same time confirms the accuracy of this study. The following Fig. 2 supports the idea that the two criteria we recommend to methodologists for evaluating MCDM methods should be used simultaneously. In other words, if the following Fig. 2 is carefully examined, it shows that both methods act in parallel, that is, successful, mediocre, and unsuccessful methods in both methods are similar to a certain extent.

The analysis findings of this study prove that in general (for both criteria) FUCA and CRADIS methods for financial data are stable to success, TOPSIS mediocre, VIKOR volatile, and COPRAS, CODAS, ELECTRE underperforming. Moreover, this study shows that SWARA, which is a subjective weighting technique, is clearly more successful than the CRITIC method, which is an objective technique.

### Limitations in the study

The MCDM evaluation criteria proposed here are general and can be easily used for ranking problems in any scientific field. However, the comparative results obtained are mostly valid for financial data. MCDM methods either use some normalization techniques such as Sum, Min-Max, Max, and Vector or may work without normalization like some 'outranking' methods. Since financial data is quite volatile (skewness-kurtosis is variable), periodic normalization methods do not always show the same performance. Therefore, data in other fields may result in the success of another MCDM result. For example, for financial data, FUCA may perform best, while for economic data, CODAS may perform best. Data type and structure is the most important limitation of this study. Firm data, normalization type, quarters, MCDM methods, weighting methods, and some threshold values used in this study are other limitations of this study. Although many methods and data types are used extensively in this study to overcome the limitations of the problem, the problems explained here should be considered as a limitation.

# CONCLUSIONS

Choosing the best among the multi-criteria alternatives is a classic MCDM problem in the selection of a high-performing firm, as in other areas of expertise. However, the literature states that it is a mystery as to which MCDM method should be preferred among the 200 methods. Even the smallest indirect development can be considered a very important step in this regard. Although it is a complex and insoluble problem in theory, the results of applied data analytics of financial markets show us that the situation is promising. In fact, the relationship between FP (financial performance based on MCDM) and SR (share return), which is defined as the success status of companies, promises a special and natural solution in this regard. As it is known, these two dynamic systems have simultaneous and meaningful relationships. Therefore, the success of a few MCDM methods that ensure this relationship at the highest level and sustainably cannot be a coincidence. While classical studies focus on the mathematical theoretical background of the subject, this study shows that real-life data analytics can offer solutions from a different perspective. According to the data of the manufacturers traded in BIST in Turkey, our findings show that the FUCA and CRADIS method (with these constraints and data conditions) is a more suitable method than the other five popular methods. Because these methods have a better-established relationship with real-life rankings. Moreover, the RR (rank reversal) performance of these methods (when we approach the subject for the first time with a statistical procedure proposed in this study) is better.

Unlike previous studies, very wide time periods, alternatives, and comparisons of MCDM methods and MCDM have been comprehensively discussed in the analysis without leaving any room for coincidence. This study proposes two different frameworks simultaneously for the comparison, evaluation, and selection of MCDM methods, which is its original and innovative approach. The model framework in this study assists the robustness, sensitivity, and validation analysis frequently used in the MCDM evaluation methodology. The proposal of this study can also help develop any MCDM method. Because with the help of this procedure, new and efficient MCDM algorithms can be developed. Therefore, in our opinion, our study proposes robust, original, and objective criteria for the comparison, selection, and development of MCDM methods.

Recommendations for future researchers

As practiced in this study, if researchers can find a real-life anchor or a relevant sequencing sequence, they can apply the MCDM capacity assessment model in their field of scientific expertise. Moreover, the RR criterion is more universal and can be easily used in all fields of science. In addition to the MCDM selection, algorithm, normalization, threshold, and preference function selection can also be made through the model here. Since the MCDM evaluation criteria proposed here are objective, they can also be used as a validation tool in the production of a new MCDM. Finally, we also recommend users compare fuzzy-based data and net/exact values with these criteria.

### Funding
The authors received no funding for this work.

### Competing Interests
Željko Stević is an Academic Editor for Peerj. All other authors declare that they have no competing interests.

### Author Contributions
- Mahmut Baydaş conceived and designed the experiments, performed the experiments, analyzed the data, performed the computation work, prepared figures and/or tables, and approved the final draft.
- Tevfik Eren conceived and designed the experiments, performed the experiments, prepared figures and/or tables, and approved the final draft.
- Željko Stević analyzed the data, performed the computation work, authored or reviewed drafts of the article, and approved the final draft.
- Vitomir Starčević performed the computation work, authored or reviewed drafts of the article, and approved the final draft.
- Raif Parlakkaya performed the experiments, prepared figures and/or tables, and approved the final draft.

### Data Availability
   The raw data is available in the Supplemental File.

### Supplemental Information
Supplemental information for this article can be found online at http://dx.doi.org/10.7717/peerj-cs.1350#supplemental-information.

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
