# Peer review of "Proposal for an objective binary benchmarking framework that validates each other for comparing MCDM methods through data analytics"

_PeerJ Computer Science, doi:10.7717/peerj-cs.1350_

## Round 0.1 · original submission · Major Revisions

Good work in methodical and practical terms, but there have been some issues highlighted. They must be addressed.

Reviewer 1 ·

Basic reporting

Proposal for an objective binary benchmarking framework that validates each other for comparing MCDM methods through data analytics
The paper represents a very good study with strong proof necessary to discuss such a problem. The authors have proposed a novel framework for comparing MCDM methods taking into account the financial performances of 140 companies. This is very hard work and congratulations on that.
The authors are very familiar with the fields covered in the paper. It is well written with the following advantages:
- The abstract is very concise with all standard elements. Clearly, the aims, main contributions, novelty, and verification of results are concisely described in the abstract. Even, it is slightly longer than usual.
- The overall quality of the paper.
- The paper has a good structure.
- Introduction section is extensive. The authors have ensured clearly the motivation for writing this study, and clear motivation for forming and using such methodology.
- Developing a novel framework with clear contributions.
- The authors have shown an extensive literature review with a large number of relevant studies.
- Well introducing of observed problem.

Experimental design

- Besides the literature review being very extensive, section 2.1. should be extended with more quality studies.
- It is needed to explain using SWARA and CRITIC methods to determine criteria weights. Why these methods?

Validity of the findings

No comment.

Additional comments

- In the introduction should be added a short description of the paper as the last paragraph.
- Each abbreviation should be defined in place of the first appearance. For example page 3 - PROMETHEE-2, VIKOR, or TOPSIS, SWOT...
- You should uniform decimal places. In Tables 3 and 4 are data with 6 decimal places, while the other is 3. It is enough to be 3.
- The last column of Table 7 should be three decimal places also.
- The same is for Tables 9 and 10.
- Ensure at least one paragraph with a description of continue of this research.
- It is better that appendix will be a supplementary file.

Reviewer 2 ·

Basic reporting

This article deals with an important issue - the reliability and consistency of the rankings generated using different algorithms of MCDM methods In this aspect, the development of an objective modelling framework is an important research problem. Many works have analysed the mathematical robustness of the computational procedures of MCDM methods. This paper fills the research gap well by focusing on the output dimensions of the models and, the performance in terms of rank reversal (RR). This study proposes two criteria that are mutually supportive. On the practical dimension, the study is embedded in the problem of evaluating the financial performance (FP) of 140 listed manufacturing companies using 9 different MCDM methods.
The paper is interesting from both a methodological and practical point of view. The paper is written according to rules and well positioned in journal aim and scope. Literature is up to date and relevant.
The paper may be accepted for publication if the following comments are met:
- the methodological and practical contribution should be clearly indicated in the intro section
- Figure 1 should be corrected - object descriptions should be centred
- the language layer should be improved - native speaker assistance is suggested
- drawings should be standardised
- the Conclusion section should be renamed Conclusions
- Appendix should be reformatted - I suggest removing the tabular layout for the description of MCDM methods

Experimental design

- the methodological and practical contribution should be clearly indicated in the intro section
- Appendix should be reformatted - I suggest removing the tabular layout for the description of MCDM methods

Validity of the findings

no comment

Additional comments

I found study interesting both in methodical and practical terms

Reviewer 3 ·

Basic reporting

Dear authors

The article has an interesting topic. However, there are some minor points that you may need to consider to improve the quality of your study. The article needs some more deep ideas about the problem. The methodology and analysis look great, but the problem and its related field are somehow incomplete. Also, I would like to know why you chose these MCDM methods, not others.

Experimental design

Looks complete.

Validity of the findings

Very well desigend.

Additional comments

Dear authors

Please consider my comments in the basic report part. The article can be accepted after a minor revision.

---

## Round 0.2 · accepted · Accept

The authors have addressed the reviewers' concerns and answered their questions appropriately. As a result, the article is ready for acceptance and publication.

Reviewer 1 ·

Basic reporting

No comment.

Experimental design

No comment.

Validity of the findings

No comment.

Additional comments

All the reviewers' comments have been addressed carefully and sufficiently. The revisions are rational from my point of view. I think the current version of the paper can be accepted.

Reviewer 2 ·

Basic reporting

This article deals with an important issue - the reliability and consistency of the rankings generated using different algorithms of MCDM methods In this aspect, the development of an objective modelling framework is an important research problem. Many works have analysed the mathematical robustness of the computational procedures of MCDM methods. This paper fills the research gap well by focusing on the output dimensions of the models and, the performance in terms of rank reversal (RR). This study proposes two criteria that are mutually supportive. On the practical dimension, the study is embedded in the problem of evaluating the financial performance (FP) of 140 listed manufacturing companies using 9 different MCDM methods.
The paper is interesting from both a methodological and practical point of view. The paper is written according to rules and well positioned in journal aim and scope. Literature is up to date and relevant.

I found my previous suggestions adressed thus I suggest to accept the paper

Experimental design

I found my previous suggestions adressed thus I suggest to accept the paper

Validity of the findings

I evaluate novelity and impactr of the paper high.
Data are robust and staistically sound

Additional comments

I found study interesting both in methodical and practical terms
Paper can be published

Reviewer 3 ·

Basic reporting

The article looks ready to be accepted.

Experimental design

Had some progress.

Validity of the findings

Great.

Additional comments

Dear authors

The article can be accepted in its current form.